



**A free, open-source method for automated mapping of quantitative mineralogy from energy-dispersive X-**
**ray spectroscopy scans of rock thin sections**
Miles M. Reed[1], Ken L. Ferrier[1], William O. Nachlas[1], Bil Schneider[1], Chloé Arson[2], Tingting Xu[3], Xianda
Shen[4,5], and Nicole West[6]
[1] Geoscience, University of Wisconsin-Madison, United States
[2] Civil and Environmental Engineering, Cornell University, United States
[3] Hopkins Extreme Materials Institute, John Hopkins University, United States
[4] Key Laboratory of Geotechnical and Underground Engineering of Ministry of Education, Tongji University,
China
[5] Department of Geotechnical Engineering, Tongji University, China
[6] Independent Researcher
*Correspondence to*: Miles Reed (miles.reed@wisc.edu)
**Abstract**
Quantitative mapping of minerals in rock thin sections delivers data on mineral abundance, size, and spatial
arrangement that are useful for many geoscience and engineering disciplines. Although automated methods for
mapping mineralogy exist, these are often expensive, associated with proprietary software, or require
programming skills, which limits their usage. Here we present a free, open-source method for automated
mineralogy mapping from energy dispersive spectroscopy (EDS) scans of rock thin sections. This method uses a
random forest machine learning image classification algorithm within the QGIS geographic information system
and Orfeo Toolbox, which are both free and open source. To demonstrate the utility of this method, we apply it to
14 rock thin sections from the well-studied Rio Blanco tonalite lithology of Puerto Rico. Measurements of
mineral abundance inferred from our method compare favourably to previous measurements of mineral
abundance inferred from X-ray diffraction and point counts on thin sections. The model-generated mineral maps
agree with independent, manually-delineated mineral maps at a mean rate of 95%, with accuracies as high as
96% for the most abundant phase (plagioclase) and as low as 72% for the least abundant phase (apatite) in these
samples. We show that the default random forest hyperparameters in Orfeo Toolbox yielded high accuracy in the
model-generated mineral maps, and we demonstrate how users can determine the sensitivity of the mineral maps
to hyperparameter values and input features. These results show that this method can be used to generate accurate
maps of major mineral phases in rock thin sections using entirely free and open-source applications.
**1 Introduction**

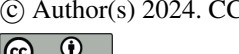


Minerals are the fundamental units of rocks and many engineered materials (Perkins, 2020; Callister and
Rethwisch, 2020). Improving the quantification of mineral properties is a longstanding research objective in
industry and academic research (Pirrie and Rollinson, 2011), given the importance of mineral properties in
chemical weathering (e.g., Hilton and West, 2020), rock damage (e.g., Shen et al., 2019; Xu et al., 2022),
planetary evolution (e.g., Hazen et al., 2008), crustal deformation (e.g., Burgmann and Dresen, 2008), and
nutrient supply (e.g., Callahan et al., 2022). Quantitative mineralogy, the mapping of mineral phases across a
sample, results in measurements of mineral modal abundance, mineral grain size and shape, and the spatial
arrangement of minerals amongst one another (Schulz et al., 2020). Modal abundance is useful because it can
yield information on the sedimentary and tectonic environments in which the rock formed (Harlov et al., 1998;
Hupp and Donovan, 2018), while the spatial arrangement of minerals in a rock, termed rock fabric, can yield
further data on mechanical anisotropy and paleo-environmental conditions during the rock's formation and
metamorphism (Přikryl, 2006; Bjørlykke, 2014). Simultaneous quantification of modal mineralogy and detailed
mapping of the spatial arrangement of minerals in an automated manner, or automated mineralogy, is thus a key
tool for investigating many geologic processes. Wide adoption of automated mineralogy techniques are limited
by the prohibitive cost or programming skills required to use many automated mineralogy software applications,
so this technique has been mostly restricted to ore characterization, resource processing, and petroleum geology
(Nikonow and Rammlmair, 2017; Schulz et al., 2020).

In practice, automated mineralogy methods use a combination of image analysis and classification methods to
identify mineral phases from elemental composition data (or their derivatives), which can be collected with a
variety of analytical methods, including energy dispersive X-ray spectroscopy (EDS), wavelength dispersive X-
ray spectrometry (WDS), micro-X-ray fluorescence (μ-XRF), and laser-ablation inductively-coupled mass
spectroscopy (LA-ICP-MS) (Nikonow et al., 2019). Automated mineralogy is being slowly adopted by
researchers outside of resource extraction for combined modal analysis of bulk mineralogy, estimates of grain
size distribution, and mineral association (Han et al., 2022), which can be useful in a variety of disciplines such
as petrology, applied geochemistry, and rock mechanics (Sajid et al., 2016; Elghali et al., 2018; Rafiei et al.,

58  2020).


Automated mineralogy from EDS with the aid of back-scattered electron (BSE) imaging has been developing
since the 1980s and has grown alongside advances in scanning electron microscopy (SEM) and image processing
algorithms (Miller et al., 1983; Fandrich et al., 2007). Commercial automated mineralogy systems are available



as integrated hardware-software systems or as standalone software packages which are combined with scanning
electron microscopes (Schulz et al., 2020). Some systems only work with certain scanning electron microscopes
and detectors from the same company QEMSCAN (Gottlieb et al., 2000), FEI-MLA (Fandrich et al., 2007), and
TESCAN TIMA-X (Hrstka et al., 2018). Others are purely software-based solutions which are integrated with
various SEMs: ZEISS Mineralogic, Oxford AZTecMineral, and Thermo-Scientific MAPS Mineralogy. The price
of hardware and software upgrades required to accommodate these systems renders them cost prohibitive to
many labs outside the resource extraction industry (Nikonow and Rammlmair, 2017). All systems have some
general ability to classify EDS spectra based on a database of pre-defined and/or customizable mineral spectra
standards (Schulz et al., 2020). Since the underlying software is proprietary, no source code is available for these
systems, and details on how they use spectra to classify mineral phases are sparse to non-existent (Kuelen et al.,
2020). Furthermore, the accuracy of mineral phase prediction from these systems has rarely been quantified
(Blannin et al., 2021).

To date, several open-source (i.e., source code is available and modifiable) automated mineralogy solutions have
been implemented. Ortolano et al. (2014, 2018) predicted mineralogy from a multistep workflow involving
principal component analysis, maximum likelihood classification, and multi-linear regression performed on EDS
or WDS spectral data using the Python extension within ArcGIS. Li et al. (2021) used a variety of legacy
machine-learning and deep-learning models to classify minerals in oil reservoir rocks using mineral maps
generated from proprietary software as training data. In terms of image classification, deep-learning methods are
state of the art but currently require the user to be relatively adept at programming and knowledgeable of the
computer vision principles employed (Khan et al., 2018; Zhang et al., 2019). A method that requires little to no
programming ability would allow more users to benefit from automated mineralogy data. An example of this
approach is XMapTools by Lanari et al. (2014), a graphical, open-source automated mineralogy solution with
multiple machine-learning classification algorithms within a standalone, MATLAB-based environment.

The main goal of this study is to present a new, user-friendly quantitative automated mineralogy method that we
developed and implemented within QGIS, a free and open-source geographic information system. Nikonow and
Rammlmair (2017) previously showed success in adapting the proprietary remote-sensing package ENVI to do
automated mineralogy using μ-XRF data. Here, we use the free and open-source Orfeo Toolbox plugin for QGIS
(Grizonnet et al., 2017) to predict thin-section scale bulk mineralogy from EDS elemental intensity data using a
random forest (RF) image classifier (Breiman, 2001). Random forest classification is a supervised classification



algorithm (i.e., the user generates training data) in which an ensemble of decision trees produces a majority vote
that assigns a thematic classification to unknown data (Breiman, 2001). Each decision tree within the ensemble is
trained on a random sampling of the training data using only a set number of random features at each branch
(Cutler et al., 2011). During prediction, for each decision tree, unknown data traverses a sequence of rule-based
branches which culminate in the assignation of a predicted class (Breiman, 2001). Each tree gets one vote for
each pixel; the predicted class with the most votes is assigned to the unknown data. There are several reasons
why RF classification is useful for automated mineralogy mapping. It is well suited for accommodating
unbalanced training data and nonparametric data distributions (Maxwell et al., 2018), which are common in rock
samples due to large differences in relative mineral abundances and elemental intensities (Ahrens, 1954). In
addition, recent work showed that RF classification performed better than other legacy machine-learning
algorithms (e.g., support-vector machines; Hearst et al., 1998) in mineral classification of reservoir rocks (Li et
al., 2021).

Unlike previous methods, the method presented here uses only freely available and open-source applications, and
it requires no programming on the part of the user. Situating the workflow within a GIS environment has
advantages over standalone programs such as direct access to raster and vector manipulation and analysis tools
and database management (Tarquini and Favalli, 2010; Berrezueta et al., 2019). In the remainder of this study, we
present an overview of the automated mineralogy method and apply it to a set of rock samples from the Rio
Blanco tonalite to demonstrate the method's utility. By outlining an easy-to-use and open-source solution, we
hope this method provides a tool for automated mineralogy to a broader community of users.

**2 Overview of the method**

The goal of our automated mineralogy method is to produce quantitative mineralogy maps of rock thin sections
solely from EDS data. Here in Section 2, we briefly summarize each step needed to reach a predicted mineral
map. In Section 3, we demonstrate how to use the method by applying it to a set of rock thin sections, during
which we elaborate on the choices users need to make and the functions they need to use during each step. We
also provide a detailed step-by-step guide in the supplementary information (Reed et al., 2024).

The starting point for this method is EDS-generated scans of rock thin sections. For the purposes of our method,
we take these scans as already measured and in hand. Generating such scans requires preparing thin sections and



analyzing them with a scanning electron microscope, both of which are done by established procedures
(Goldstein et al., 2018). The necessary output from such scans are rasters of elemental intensity (counts/eV), one
for each element of interest (e.g., Ca, Na, K, etc.). After the EDS elemental intensity rasters have been generated,
all the remaining steps in the method are conducted in QGIS. No programming is required in any step. Instead,
users need only be familiar with QGIS and their samples.

The first step is to ensure all the necessary information is in place. This involves importing the raw elemental
intensity rasters into QGIS with no coordinate reference system (Fig. 1a). This also involves compiling a list of
all the mineral phases that will be mapped in the thin section, which can be assessed based on prior knowledge,
literature, and examination of EDS spectra. As we describe in Section 4, we recommend restricting this to
mineral phases with sufficiently high abundance to be accurately mapped.

The second step is to smooth the raw elemental intensity rasters (Fig. 1b. This is useful because EDS-generated
elemental intensity rasters are subject to noise, which can arise through electron beam interactions with the
sample and incorrect spectral peak identification by the EDS software (Goldstein et al., 2018). As we describe in
Section 4.3, we found that this smoothing step was best done with a 7-pixel radius circular mean filter. Here, a
mean filter is an image processing operation where a circular sliding window with a fixed radius surrounding a
center pixel moves across an input raster one pixel at time and, in an output raster, assigns a mean value to the
center pixel based on the surrounding pixels (Gonzalez and Woods, 2017). We performed this on intensity rasters
from the example samples we applied our method to in Section 3. For this, we used the free and open-source
System for Automated Geoscientific Analyses (SAGA) plugin for QGIS (Conrad et al., 2015).

The third step is to gather the smoothed elemental intensity rasters into a virtual raster with one band for each
element of interest (Fig. 1c). For example, if the user chooses to import elemental intensity rasters for six
elements, as we did in the application of this method to our samples in Section 3, this will result in a virtual raster
with six bands. For this, we used the Geospatial Data Abstraction Library (GDAL/OGR contributors, 2022),
which is a standard library in QGIS.

The fourth step is to train a RF image classification model on the virtual raster (Fig. 1d). This requires generating
a large number (~hundreds) of small polygons on the virtual raster. Each of these small polygons must lie within
a single mineral phase, which the user must identify and assign to the polygon. Collectively, these small polygons





must cover all the mineral phases of interest in the thin section in sufficient number to train the RF model. If the
user wishes to assess the accuracy of the RF-predicted mineral map to a manually mapped portion of the thin
section after the method is complete, we recommend restricting the location of these small training polygons to a
relatively small portion of the thin section (~10-20%). This will ensure that other portions of the thin section can
be mapped manually to compare against the RF-predicted mineral map. If the user does not wish to conduct such
an accuracy assessment after the RF-predicted mineral map is complete, then these small training polygons can
be generated anywhere across the entire thin section.

After the RF model has been trained, the fifth step is to apply the trained RF model to the entire virtual raster
(Fig. 1e). During this step, the RF model assigns a mineral phase to every pixel in the virtual raster, which yields
a mineral map for the entire thin section. For these RF modeling steps, we used the free, open-source Orfeo
Toolbox plugin for QGIS (Grizonnet et al., 2017).

The sixth and final step is to denoise the RF-generated mineral map (Fig. 1f). For this, we applied a circular
majority filter using the SAGA plugin for QGIS. A majority filter is akin to the mean filter described above but
assigns the modal value of the surrounding pixels to the pixel in the output raster at the center sliding window
(Gonzalez and Woods, 2017). As we describe in Section 4.3, we found that this was best done with a 10-pixel
radius majority filter in the example samples we applied this to in Section 3. This eliminates most isolated pixels
within larger groups of pixels of a uniform predicted mineral phase and rare pixels that were not classified due to
voting ties (Ortolano et al., 2018; Nikonow et al., 2019)

At this stage, the RF-predicted mineral map is complete. It can now be interrogated or manipulated according to
the user's needs. For instance, the mineral map can be converted from a raster to a vector form to facilitate
measurement of mineral grain size and other properties (Section 5.2).



**Step 1: Import elemental intensity rasters**

**Step 5: Create mineral map**

**Step 2: Smooth elemental intensity rasters**

**Step 6: Smooth mineral map**

**Step 3: Create virtual raster**

**Step 4: Train random forest model**





**Figure 1**. Example application of the automated mineralogy method. (a) Step 1: Import raw elemental intensity rasters (Ca, Na, Mg, Fe, K and Ti) into QGIS. Here, the rasters shown are for the thin-section sample 1-13a. The zoomed-in view of the Ca raster exemplifies the short-wavelength noise in the elemental rasters. (b) Step 2: Smooth each elemental intensity raster with a circular mean filter. The zoomed-in view shows that this filter has eliminated much of the short-wavelength noise that was in the raw elemental rasters. (c) Step 3: Create a virtual raster by combining the smoothed elemental rasters into a single image container with bands for each element. The white circle shows the area within which polygons were generated to train the random forest (RF) model in Step 4. (d) Step 4: Within the training area boundary in the virtual raster (large white circle, as in Step 3), draw a series of small polygons (here, small white circles). Each polygon must lie within a single known mineral phase, and collectively these small polygons must sample all mineral phases of interest (here, plagioclase feldspar, quartz, hornblende, biotite, potassium feldspar, Fe-Ti oxides, apatite, and chlorite). These polygons collect the pixel-level data on which the RF model will be trained. (e) Step 5: Apply the trained RF model to the entire sample to create a thin section-scale mineral map. (f) Step 6: Smooth the RF-predicted mineral map with a circular majority filter.



## 3 Application of the method


### 3.1 Preparation of rock thin sections from the Luquillo Critical Zone Observatory

To demonstrate the utility of the method described in Section 2, we applied it to 14 thin sections of Rio Blanco tonalite from the Luquillo Critical Zone Observatory (LCZO) in Puerto Rico, United States, a site that has been the subject of substantial research on the weathering of igneous rocks into saprolite and soil (White et al., 1998; Riebe et al., 2003; Stallard and Murphy, 2012; Brocard et al., 2023). The lithology is a phaneritic, plutonic igneous rock with some evidence of low-grade hydrothermal alteration (Speer, 1984). The Rio Blanco tonalite provides an ideal case study because mineral abundance has been characterized previously via quantitative X-ray diffraction (XRD) and point counting modal analysis (i.e., systematic manual identification and counting under microscope; Ingersoll et al., 1984), which indicated the rock consists of plagioclase feldspar (andesine), quartz, biotite, hornblende, potassium feldspar, magnetite, apatite, and chlorite (Murphy et al., 1998; Buss et al., 2008; Ferrier et al., 2010).




To ready the samples for EDS, 14 petrographic thin sections were prepared on 27 x 46 mm glass slides from
bedrock core quarters collected from the Rio Icacos catchment within the LCZO (Comas et al., 2019). The
samples ranged in area from 34.7 to 139.5 mm$^2$. Four samples are composed of weathered rock nearer to the
surface while the rest are more pristine bedrock (Orlando et al., 2016). From each core depth, two thin sections
were prepared in vertical and horizontal orientations. Our own preliminary optical microscopy observations
revealed that these samples contained abundant plagioclase, quartz, hornblende, and biotite, which is consistent
with previous modal analyses (Murphy et al., 1998; Buss et al., 2008).

**205  3.2 Measuring elemental intensity in thin sections with energy dispersive spectroscopy**

Each thin section was mapped with energy dispersive X-ray spectroscopy (EDS) using a Hitachi S-3400 VP-
SEM with a thermionic tungsten electron source equipped with an Oxford Instruments X-Act 10 mm$^2$ silicon drift
detector receiving X-rays across 2048 spectral bands. The EDS detector acquires a spectrum showing the energy
and intensity of characteristic X-rays emitted from the sample to determine the atomic composition of the sample
within the analysis volume of the primary beam (Goldstein et al., 2018). For our measurements on our thin
sections, the beam step size and magnification resulted in full thin-section elemental intensity maps (counts/eV)
at a resolution of 4 μm/pixel. EDS data were acquired with accelerating voltage of 15 kV and beam current of
~10 nA. Dwell time per beam step, which governs the amount of time the detector counts X-rays, was 200 ms
(Newberry and Ritchie, 2013a). EDS acquisition time was ~3.5 hours for each thin section.

From the EDS analysis application included with this instrument (AZtec), we exported six TIF files for each
sample (Fig. 1a) consisting of full-resolution elemental intensity rasters for the elements of interest (Ca, Na, K,
Mg, Fe and Ti). These rasters contain the X-ray counts of elemental intensity at each pixel and have a mean size
of over 20 megapixels over the 14 study thin sections. We selected these elements because they are present in
varying abundance in the mineral phases within the Rio Blanco tonalite, and hence are useful for distinguishing
among the mineral phases in these samples. For example, K, Mg, Fe, and Ti are present at high abundance in
biotite (Dong et al., 1999) but are present at low abundance in other major mineral phases in this lithology (e.g.,
plagioclase feldspar, quartz). Our initial attempts at classification showed that the inclusion of rasters of Si and Al
had no effect on classification accuracy, so we did not include them here.



This method requires a list of mineral phases present in the samples for both training of and prediction by the RF
models (Steps 4 and 5 in Section 2). Such a list can be obtained in a variety of ways, including prior studies of
qualitative mineralogy of the host lithology or mineral identification from optical microscopy on the sample thin
sections. For the 14 samples analyzed here, we generated a list of mineral phases by inspecting the EDS-
generated X-ray spectral data within Oxford AZtec, a proprietary software package integrated with the SEM that
we used to measure EDS scans of our samples. From these spectra we identified plagioclase feldspar, quartz,
hornblende, biotite, potassium feldspar, Fe-Ti oxides (predominantly magnetite-titanomagnetite), and apatite as
mineral phase classes for the RF models (Section 3.3). For those without offline access to a full EDS
environment, some systems such as Oxford AZtec allow for the full export of data into text or binary formats,
which can be accessed with free and open-source tools (e.g., HDFView or NIST DTSA-II). Due to trace
abundance (Murphy et al., 1998), other minerals present in the samples like epidote and titanite lacked an
adequate number of trainable examples, so were neglected or combined with an associated phase, Fe-Ti,
respectively. For reference, the mean abundance of apatite, the lowest abundance mineral phase we trained, was
~0.1%. We recommend that phases present at abundances lower than this be omitted or combined.

**3.3 Smoothing and virtualization of the elemental intensity rasters**

We smoothed each elemental intensity raster with a 7-pixel radius circular mean filter using SAGA's Simple
Filter tool to eliminate noise in the EDS data. We chose this filter size because it optimized the accuracy
calculated during the training and validation of the RF model. We test the sensitivity of this choice in Section 4.3.
We then used the GDAL gdalbuildvrt command within QGIS to group the smoothed elemental intensity rasters
into a virtual raster dataset, in which each elemental raster is represented as a separate band. A virtual raster is a
container for multiple rasters that encodes metadata such as file locations and other attributes in extended markup
language (XML) (McInerney and Kempeneers, 2014). Opening and processing virtual raster datasets requires
less computer resources as the underlying rasters are only accessed when required.

**3.4 Training random forest models for mineral classification**

Before a RF model can be tasked with assigning a mineral phase to every pixel in an entire thin section, it must
first be trained upon the mineral phases in the thin section. On each of the virtual rasters for the 14 thin sections,
we selected an area encompassing less than ~15% of the total thin-section sample area within which we trained
the model. We selected training areas that represented all mineral phases as well as possible, so that each phase
would receive an adequate amount of training data for each phase. Selecting a small training area in the thin



section is useful because it enables users to test the accuracy of the trained model on other areas of the thin
section, if desired. This is not a necessary step in the method, but in Section 4 we show how such accuracy tests
can be done on other portions of the thin sections.

For each mineral phase within the training area, we manually generated hundreds of circular polygons upon the
virtual raster using the knowledge gained previously from examining the EDS spectra (Fig. 1). A single training
polygon within the training area collects all pixel values contained within it from each elemental intensity raster
composing the virtual raster. This polygon is then labelled as a single mineral phase, effectively labelling every
pixel value contained within it to that mineral phase. For a few thin sections, multiple subareas composed the
training area to incorporate enough data on less abundant minerals like apatite. Because each training polygon
encompassed pixel-level data for all bands from the virtual raster, the training datasets were large ($>10^5$ pixel-
level samples for each thin section). Training samples per mineral phase were highly unbalanced (i.e., some
mineral phases covered many more pixels than others) due to the high abundances of quartz and plagioclase
relative to those of minor mineral phases like apatite. Orfeo Toolbox handles this potential problem automatically
by randomly selecting samples at a rate relative to the size of the smallest class, ensuring that the minority classes
like apatite have an equal probability of being drawn into a sample subset used to construct an individual decision
tree.

Using the training data obtained from the virtual raster for each thin section, we trained RF image classification
models using the TrainImagesClassifier function in Orfeo Toolbox. In this function, users must select
hyperparameter values for the RF model. In machine learning, hyperparameters define the general behaviour of a
model, and are distinct from model parameters, which are learned through training. For more details about RF
machine learning models hyperparameters, see the review in Probst et al. (2019). We used the default
hyperparameter values (Table 1) for the models employed for our final predicted mineral maps.

A measure of model accuracy is automatically calculated by the TrainImagesClassifier function at this step using
unseen training data, which can be useful to examine before proceeding as to ensure that the RF model is
operating correctly. The accuracy metric we focus on in this study is the F1 score (Equation 3), which is the
harmonic mean of the precision metric (Equation 1) and the recall metric (Equation 2). This is a useful measure
of the accuracy of RF-predicted mineralogy because it penalizes false positives and false negatives while





rewarding true positives and neglecting true negatives (Chinchor and Sundheim, 1993), which can be very
plentiful for low abundance phases.

$$Precision = \frac{True\ positives}{True\ positives + False\ positives} \tag{1}$$


$$Recall = \frac{True\ positives}{True\ positives + False\ negatives} \tag{2}$$


$$F1\ score = \frac{2(Precision)(Recall)}{Precision + Recall} \tag{3}$$


In the application of Equations 1-3 to mineral maps, a true positive is defined as pixel-level agreement on the
presence of a given mineral phase between the model prediction and unused training data, which the algorithm
holds out from training for the purpose of calculating metrics such as the F1 score. Similarly, a true negative is
agreement on the absence of a given mineral phase. False positives and false negatives are disagreements on the
presence and absence of a given mineral phase, respectively. Application of the default hyperparameters to our
samples yielded very high F1 scores (~0.99). This gave us confidence that the predicted mineral maps generated
using the default hyperparameters were near optimal for comparison with manually delineated test maps
(described in Section 4.1).


**Table 1**. Default hyperparameter values for Orfeo Toolbox RF machine learning
model.

| Parameter name | Value |
|---|---|
| Maximum number of trees in the forest | 100 |
| Maximum depth of tree | 5 |
| Size of the randomly selected subset of features at each tree node | (number of features)$^{1/2}$ |
| Minimum number of samples at each node | 10 |







We applied each trained model to its corresponding virtual raster to predict a single mineral phase at each pixel,
except in the case of ensemble voting ties, in which case no phase was assigned to that pixel. This resulted in
mineral maps at the same resolution as the virtual rasters (~4 μm).

**309 3.5 Using the random forest models to generate mineral maps**

In our application of the trained RF models to our thin sections, the models calculated the entire thin-section scale
mineral maps in a under a minute using a desktop computer (4 GHz processor; 64 GB memory). Figure 1 shows
an example of one of these mineral maps.

After a thin section's mineral map has been generated, it is trivial to calculate the abundance of each mineral
phase by counting pixels. Figure 2 shows the abundance of each mineral phase across all 14 samples with the
error given by the mean F1 scores of the minerals. It also reveals relatively little variation in each mineral phase's
abundance among the 14 samples, which is consistent with previous observations of the Rio Blanco tonalite. The
RF-predicted mineral abundances compare well with those measured from modal analysis via point counting on
BSE imagery (Buss et al., 2008) and via quantitative XRD (Ferrier et al., 2010). Buss et al. (2008) measured
average areal abundances of 19.9% and 49.3% for quartz and plagioclase, respectively, comparable to the RF-
predicted average abundances of 22.8 ± 1.0% and 55.8 ± 2.3% (± error from mean F1 scores) on our 14 thin
sections. The combined abundance of hornblende and biotite ('Fe-silicates') measured by Buss et al. (2008) was
24%, which is close to the maximum RF-predicted abundance of 'Fe-silicates' among our 14 samples (25.0 ±
1.5%). Using common values for molar masses (M mol$^{-1}$) and densities (M L$^{-3}$), the XRD-based abundances
(converted to areal abundance) from Ferrier et al. (2010) for quartz, plagioclase, and hornblende were 24%, 62%,
and 14%, respectively, while the RF-predicted mineral maps yielded 22.8 ± 1.0%, 55.8 ± 2.3%, and 10.4 ± 0.7%,
respectively. When quartz, plagioclase, and alkali feldspar abundances are normalized for usage with a Quartz-
Alkali Feldspar-Plagioclase-Feldspathoid diagram (Le Maitre, 2002), the RF-predicted abundances for each
mineral phase demonstrated that all samples can be classified as tonalite, matching the name of the lithology.






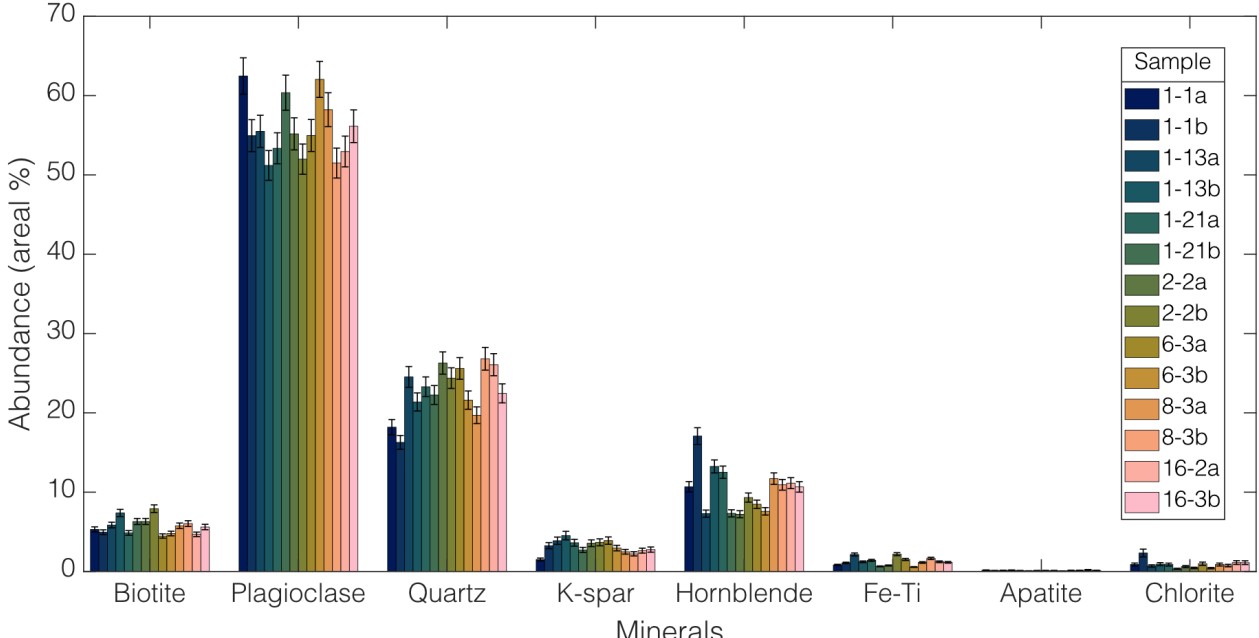

**Figure 2**. Areal abundance for all 14 samples of the Rio Blanco tonalite. Error bars stem from mean F1 scores for each individual mineral phase from test map comparisons (see Section 4.1).


**4. Accuracy of random forest-predicted mineral maps and sensitivity analyses**

**4.1 Accuracy of random forest-predicted mineral maps**
Before applying the trained RF models to the full thin sections, we manually mapped the mineralogy of a small
section for three samples (6-3a, 16-2a, and 1-13a) to assess the accuracy of the model-generated mineral maps.
We refer to these manually delineated mineral maps as "test maps". These test maps were manually delineated as
vector polygons for all mineral phases using the elemental intensity rasters for guidance. For example, when
mapping a grain of potassium feldspar, we determined the boundaries of the grain with filtered and unfiltered
rasters of K as well as combined intensity rasters of multiple elements. We consider these maps to be 'ground
truth' data, which are never perfect representations of reality (Foody, 2024), but, nonetheless, may serve to
compare the performance of this method to the extremely slow process of manually mapping grain boundaries.
We then rasterized the manually-delineated vector maps, which resulted in the classification of every pixel within
the test maps as one of the eight mineral phases. The test maps averaged over 1 million pixels in size.





We compared the same section of the predicted mineral maps to the test maps using a frequency-weighted F1
score (Equation 4) to gauge the average accuracy for all mineral phases. To calculate a frequency-weighted F1
score, the F1 score for the $i^{th}$ class (F1 score$_i$) is weighted by the class frequency ($w_i$), which is the proportion of
pixels of class $i$ to the total number of pixels in the test map. Here, $N$ is the number of mineral phases.

$$Frequency-weighted\,F1\,score = \sum_{i=1}^{N} w_i F1\,score_i \tag{4}$$


We clipped the portion of the predicted mineral map overlapping the test map from the full map for each of the
three thin sections with a test map. From these two rasters, we calculated the frequency-weighted F1 score.

How accurate were the RF-generated mineral maps in Section 3? For the three thin sections that were mapped
both manually and by the RF-based method in Section 2, the mean frequency-weighted F1 score among the three
thin sections was 0.948 ± 0.002, meaning that nearly 95% of the pixels in the RF-predicted maps agreed with
those in the manually delineated maps (Table 2). The accuracy varied among mineral phases. The four most
abundant mineral phases (plagioclase, quartz, hornblende, and biotite) all have mean F1 scores of 0.94 to 0.96,
while apatite, the least abundant mineral phase, had the lowest mean F1 score of 0.72. A closer look at the
precision and recall metrics for apatite show that mean recall scores (0.62) were lower than mean precision
(0.91). This indicates that the models correctly predicted apatite when attempted but the models often neglected
to predict apatite. Abundance and the mean F1 score of a phase were not always linked; for example, Fe-Ti
oxides were low in abundance (~1%) but registered a mean F1 score of 0.91.

Figure 3 shows an example of an RF-predicted mineral map with misclassified pixels shown in red. This
illustrates a key point: the accuracy of the RF-predicted mineral maps is not spatially uniform. Most pixels that
diverge from manual classification occur at grain boundaries where elemental compositions shift abruptly in
space. By contrast, in mineral grain interiors, divergent pixels are far less common. This indicates that the
accuracy of RF-predicted mineralogy in grain interiors is higher than the F1 scores in Table 2.





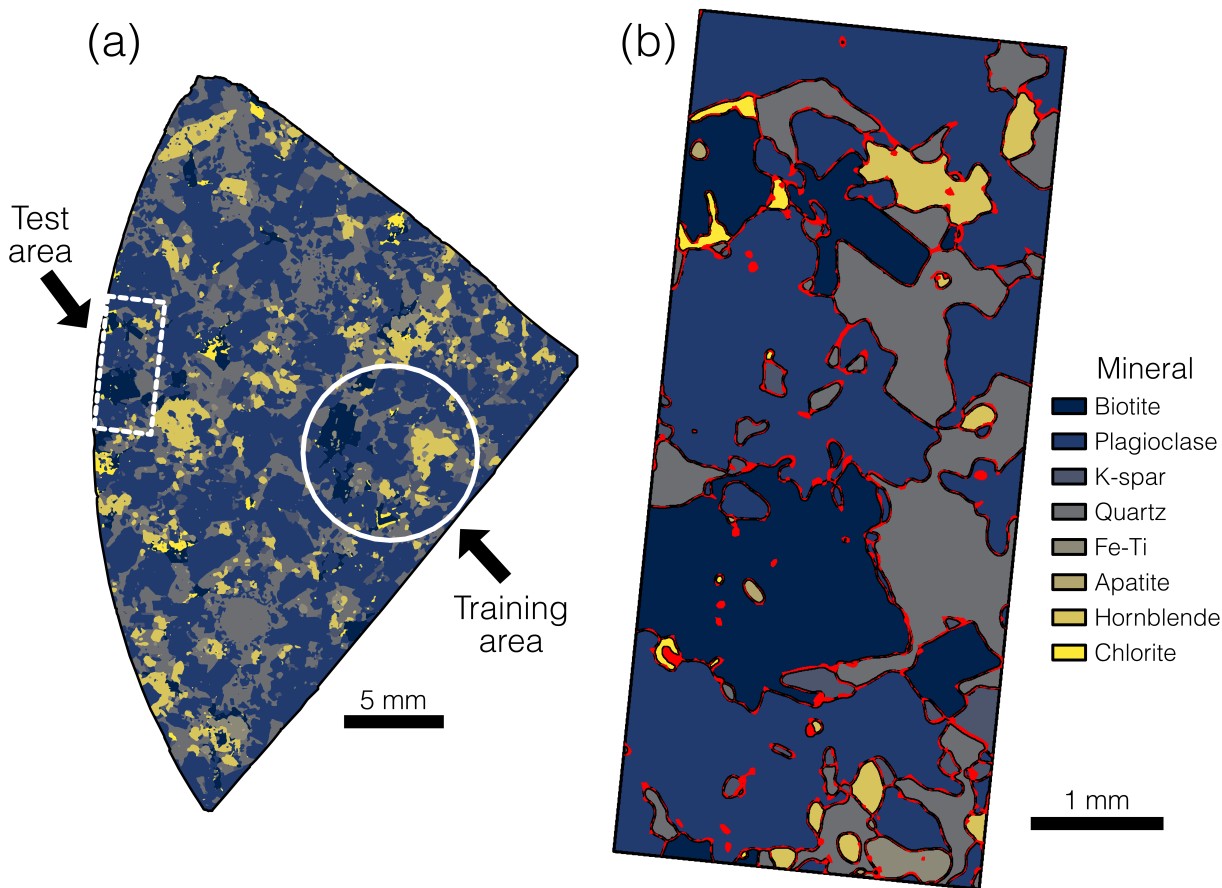

**Figure 3.** (a) Predicted mineral map for sample 6-3a, showing the location of the manually delineated test map, which we used to check accuracy. (b) Predicted mineral map for the test area. Red color signifies where pixels in the predicted map diverge from the manually delineated test map. This shows that most divergent pixels are at mineral grain boundaries.


A combined confusion matrix for pixel-level comparisons from every test and predicted map showed the most
common divergent classification was chlorite for biotite. This is likely because biotite and chlorite have similar
elemental compositions and because they often share a grain boundary (chlorite is a product of hydrothermal
alteration of biotite), which means they are more prone to disagreement along grain boundaries. Among the major
minerals, our models divergently classified potassium feldspar as plagioclase feldspar most often, likely because
many potassium feldspar grains in the Rio Blanco tonalite contain small amounts of Na, like plagioclase.





Figure 4 shows close agreement between the RF-predicted abundance and the manually mapped abundance in the
test areas, with a mean difference for a given phase of $0.45 \pm 0.02\%$ across the three test maps. So, although some
predicted pixels were misaligned spatially, the RF-predicted mineral abundances agree well with manual
estimates derived from the test maps.

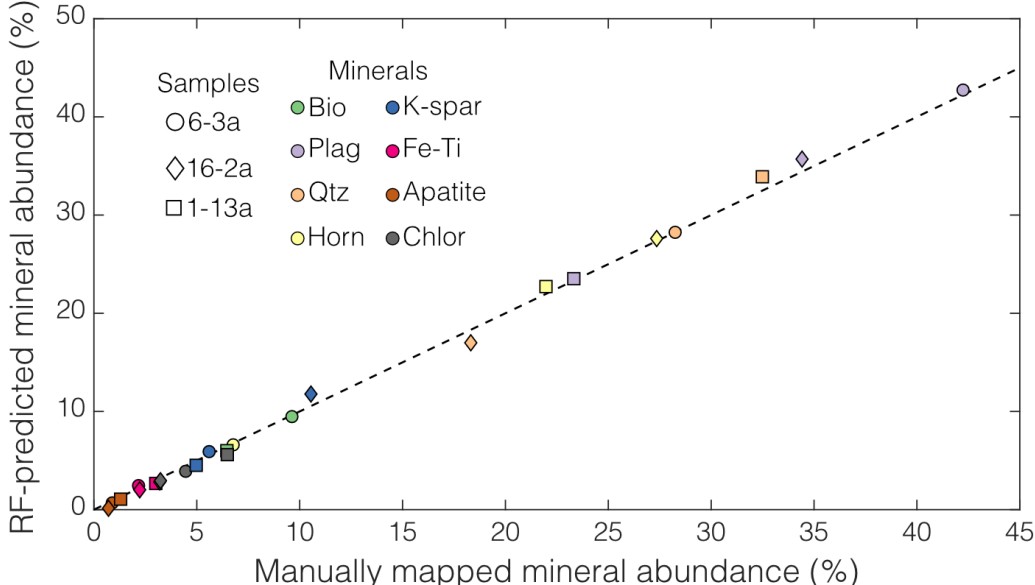

**Figure 4.** RF model-predicted mineral abundance vs. manually mapped mineral abundance in the test areas of the three samples with test maps. The dashed line is a 1:1 line. Although there was some spatial mismatch around the edge of mineral grains (e.g., Fig. 3), the RF-predicted modal abundances agree well with abundances inferred from manual mapping in the test areas.



**Table 2.** Mean F1 scores (accuracy metric) for mineral phases among the three test maps (Fig. 4), based on comparison of automated mineralogy maps to manually delineated mineralogy maps.

| Mineral | Mean F1 score |
|---|---|
| All phases (frequency-weighted) | 0.95 |
| Plagioclase feldspar | 0.96 |
| Quartz | 0.94 |
| Hornblende | 0.94 |



| | |
|---|---|
| Biotite | 0.94 |
| Potassium feldspar | 0.88 |
| Fe-Ti oxides | 0.91 |
| Chlorite | 0.79 |
| Apatite | 0.72 |


**4.2 Sensitivity of mineral maps to random forest hyperparameters and input features**

In our application of the method in Section 2 to the 14 samples in Section 3, we used a set of default values for
three RF hyperparameters: maximum tree depth, number of trees, and minimum sample size per node. Reviews
of hyperparameter tuning on RF models have shown that the number of trees and the minimum number of classes
per node can have a large effect on classification accuracy (Probst et al., 2019). How sensitive are the output
mineral maps to the user's choice of these hyperparameter values?

Orfeo Toolbox does not contain a facility for hyperparameter tuning in QGIS, so we developed a workflow to
undertake our own hyperparameter optimization outside of QGIS in Python. This is not a necessary step in the
method, but we have included this code in the Supplement for users who wish to conduct their own
hyperparameter optimization. We began by converting the smoothed elemental intensity image data in the three
training areas within the manually delineated test maps into NumPy arrays (Harris et al., 2020) using a
combination of three Python libraries: rasterio (Gillies et al., 2019), geopandas (Jordahl et al., 2020), and shapely
(Gillies et al., 2022). We then used the implementation of the RF classifier from the machine-learning package
scikit-learn (Predregosa et al., 2011) for both hyperparameter optimization using a randomized five-fold cross
validation (Breiman and Spector, 1992) and derivation of feature importance using permutation testing (Breiman,
2001). Through these operations we seek to find optimal hyperparameters and test the importance of input
features (here, elements), respectively.

We used the scikit-learn RandomizedGridCV function to systematically test the sensitivity of the output mineral
maps to the RF hyperparameter values. To do this, we trained 100 unique RF models across a range of maximum
tree depth (1-100), number of trees (10-2000), and minimum sample size per node (5-25). These hyperparameters
are common between the Orfeo Toolbox and scikit-learn implementations of the RF classifier. We used five-fold
cross-validation, in which each randomly selected set of hyperparameters is used to train the same model five



times while sampling different portions of the training data (Breiman and Spector, 1992). We report the best fit
parameters and resultant accuracy in terms of the frequency-weighted F1 score upon comparison to the test maps
using these optimized parameters.

Orfeo Toolbox has not yet incorporated a capacity to derive feature importance scores. Feature importance in RF
classification is calculated by permutation testing, which is the extent to which an accuracy metric declines if a
single input feature's unused training data is randomly altered during the training process and validation process
(Breiman, 2001; Guo et al., 2011). We used the sci-kit learn function permutation_importance to assess
importance using the frequency-weighted F1 score. We report the feature importance for the three samples with
manually delineated test maps and discuss their implications.

Tuning the hyperparameters in scikit-learn showed that both a higher maximum tree depth and number of trees
may be optimal for our RF models, while the minimum sample for splitting was more variable (Table 3). Using
these optimized RF hyperparameters within Orfeo Toolbox yielded a mean frequency-weighted F1 score of 0.95
when comparing the three samples with manually delineated test maps, which is the same F1 score realized by
using the default hyperparameters. As the two implementations of the RF classifier are somewhat different in
terms of available hyperparameters, the comparison is imperfect, but does provide a check to see if the default
hyperparameters could be improved upon. That an optimized set of hyperparameters delivered very little to no
increase in accuracy is unsurprising as RF models are known to perform well with little to no tuning if reasonable
hyperparameter values are initially used (Maxwell et al., 2018). Unless low F1 scores are realized during Step 4,
it is our recommendation that the default RF hyperparameters in Orfeo Toolbox be used.


**Table 3**. Optimal RF hyperparameters from five-fold cross validation performed using sci-kit learn.

| Sample | Maximum tree depth | Number of trees | Minimum sample for split |
|--------|--------------------|-----------------|--------------------------|
| 1-13a  | 73                 | 1685            | 25                       |
| 6-3a   | 94                 | 1371            | 5                        |
| 16-2a  | 73                 | 1581            | 5                        |





Feature importance, as determined through permutation testing, showed that both K and Mg were the most
important features for our scikit-learn trained models with mean decreases in accuracy based on frequency-
weighted F1 scores derived from the training and validation process on unused data of 0.29 for both elements
(Fig. 5). Ti was relatively unimportant with a very small, slightly positive value, implying it could be omitted.
Although Ti is present within biotite and Fe-Ti oxides in our samples, Ti showed little to no decrease in mean
accuracy as both biotite and Fe-Ti oxides can be classified using other elements. We tested whether our feature
importance scores were pertinent to models in Orfeo Toolbox by leaving out, in turn, K, Mg, and Ti during
training and validation process. Excluding K decreased mean F1 scores due to the degradation of potassium
feldspar, biotite, and chlorite accuracy. In contrast, omitting Mg did not decrease F1 scores, showing that a
feature importance score does not directly translate to decreased model accuracy upon omission (Cutler et al.,
2011). Leaving out Ti had little effect on F1 scores. If a user of our method is unsure whether an element could
be a truly important feature, omitting an important element from the training process by creating virtual rasters
without that element should yield a notable degradation in training F1 scores.


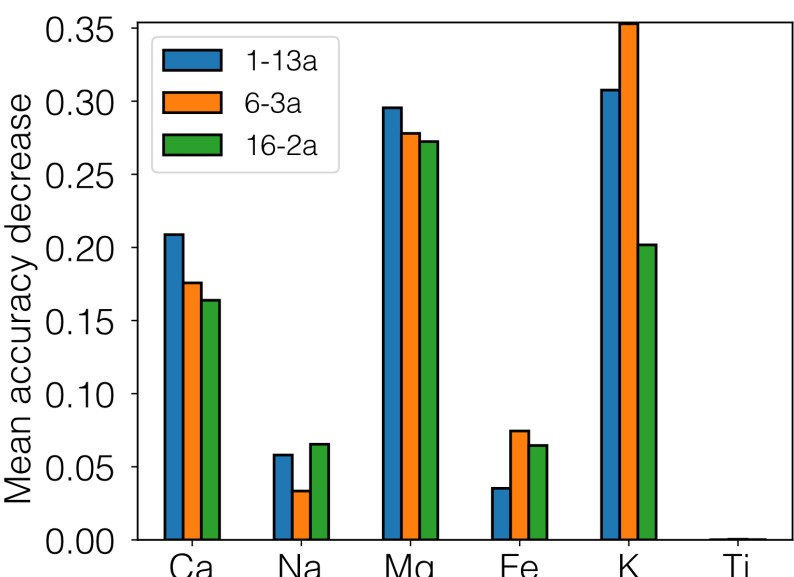

**Figure 5**. Feature importance from scikit-learn using permutation testing for all six input elements for
the three samples with test maps. Mean accuracy decrease is the change in the F1 score due to



randomly changing feature data in the unused portion of the training data during the validation process. In Orfeo Toolbox, training models that omitted K degraded F1 scores while those that omitted Mg yielded little change, indicating that feature importance score does not always directly map onto model accuracy and that some experimentation with input features (elements) during the training phase is warranted.


## 4.3 Sensitivity of mineral maps to filter sizes

In our application of this method to our samples, we applied a circular, 7-pixel radius mean filter to the EDS-
generated elemental intensity rasters (Step 2 in Section 2), and we applied a circular, 10-pixel radius majority
filter to the output mineral maps (Step 6). To quantify the sensitivity of the output mineral maps to these
"hidden" parameters, we generated a series of RF models across a range of mean filter radii for the elemental
intensity rasters (no filter, 2, 5, 7, 10, and 20 pixels) and a range of majority filter radii (no filter, 2, 5, 7, 10, and
20 pixels). For the three thin sections with manually delineated mineral maps, we calculated the frequency-
weighted F1 score of the entire thin section by comparing each of the RF-predicted mineral maps to the manually
delineated test maps.

Figure 6 reveals that both the mean filter and the majority filter affect the accuracy of the predicted mineral maps.
The largest impact on the accuracy, as measured by F1 score, was in the application of any mean filter at all to
the input elemental intensity rasters. The left panel in Fig. 6 shows that applying no mean filter to the elemental
intensity rasters produced low F1 scores (0.52-0.69) for all models and all samples, regardless of the size of the
majority filter. Accuracy increased with mean filter radius up to 5 and 7 pixels, which yielded high F1 scores at
all majority filter sizes (0.91-0.96) due to the elimination of spurious inclusions within larger mineral grains
(middle panels in Fig. 6). Beyond that size, accuracy decreased slightly with higher mean filter radius, with lower
F1 scores at radii of 10 pixels (F1 scores of 0.90-0.95) and 20 pixels (0.87-0.89). This implies an intermediate
optimal mean filter radius of 5-7 pixels for these samples.

Accuracy was sensitive to the size of the majority filter, particularly for models that applied no mean filter or a
small (2-pixel radius) mean filter to the input elemental intensity rasters (Fig. 6). For the models that applied a
mean filter of any size, accuracy was lower at small majority filter radii (0 or 2 pixels) and large radii (20 pixels)
than at intermediate majority filter radii (5-10 pixels). At the largest radii, the RF-predicted mineral grains begin
to lose shape, becoming more circular. Thus, accuracy was maximized at intermediate majority filter radii of 5-7





pixels, just as it was at intermediate mean filter radii. Excluding plagioclase and quartz (which generally do not
occur as isolated grains), the three samples with test maps (6-3a, 1-13a, and 16-2a) have a median grain area of
~0.005 mm$^2$ ($n$ = 5188 mineral grains across all three samples) while the 5-7-pixel radii filters have areas of
~0.001 mm$^2$ and ~0.002 mm$^2$, respectively. These optimal sizes most likely result from a mix of the initial EDS
pixel resolution and data quality and the types and sizes of minerals in the thin section (Lanari et al., 2014;
Ortolano et al., 2018), so we recommend that users experiment to find the optimum filter sizes for their samples.


**Figure 6.** Accuracy of the output mineral maps (as quantified by frequency-weighted mean F1 scores) for combinations of mean filter and majority filter sizes for the three samples with test maps. Each section is a single mean filter size. The most accurate mineral maps (i.e., those with the highest F1 scores) were generated using a 5- or 7-pixel radius mean filter combined with a 5- or 7-pixel radius majority filter.



**5 Discussion**

**5.1 Advantages of this open-source automated mineralogy method**
Situating our workflow in a free and open-source GIS environment confers several practical benefits. Both Orfeo
Toolbox and QGIS are frequently updated with source code that can be examined and modified, unlike many



proprietary hardware/software systems (Keulen et al., 2020). Orfeo Toolbox and QGIS each have extensive
documentation and user forums monitored by the developers, which can aid in addressing user issues (Raza and
Capretz, 2015). Incorporating open-source software into scientific methods fosters transparency and
reproducibility as the software is widely accessible and more easily scrutinized (Ramachandran et al., 2021). As
both Orfeo Toolbox and QGIS are ongoing efforts with active contributing communities, our no-code workflow
is tied to software that is not likely to fall into disrepair or unavailability, unlike much open-source scientific
software (Coelho et al., 2020). Furthermore, both Orfeo Toolbox and QGIS are available for all major operating
systems, Windows, macOS (Intel), and Linux, so this factor does not limit accessibility. Orfeo Toolbox will
likely continue to incorporate new state-of-the-art, machine-learning algorithms. For example, Orfeo Toolbox has
recently been unofficially extended to utilize the Google TensorFlow library (Abadi et al., 2016) to perform deep-
learning tasks on remote sensing imagery (Cresson, 2018, 2022). There are also efforts to develop open-source
scanning electron microscope systems and attendant software such as the NanoMi project (Malac et al., 2022).
All of this means that automated mineralogy methods are likely to become more popular and accessible.

We expect that a broad range of geoscientists will be capable of using this GIS-based method, since many
geoscience undergraduate programs incorporate GIS into courses (Marra et al., 2017). It requires no
programming skill to obtain mineral maps, thereby eliminating a potential barrier for use (Bowlick et al., 2016).
Since the workflow takes place within a GIS environment, the input elemental intensity rasters could easily be
processed in other ways besides the mean smoothing filter that we applied here, such as edge-detection filtering
or elemental intensity ratioing. Creation of optimal input features, so-called feature engineering, is fostered by the
many QGIS frontends that interface with SAGA GIS and GDAL raster manipulation programs. Our method does
not require a corresponding plugin for Orfeo Toolbox/QGIS, but much of it could be automated from the Orfeo
Toolbox/QGIS Python API or as QGIS console commands, if desired. Input parameters for image filters and
hyperparameters for the RF models can be saved as JavaScript Object Notation (JSON) files, which can be
loaded in later, overcoming some of the reproducibility issues inherent in workflows using graphical user
interfaces (Brundson, 2016).

**5.2 Illustration of the utility of random forest-generated mineral maps**
There are many potential uses for thin section-scale mineral maps once they have been generated. Converting the
mineral maps into vector form allows for the calculation of derived parameters such as median grain area for
minerals that occur as single grains (e.g., biotite), distance between grains of a mineral, and the types of minerals





surrounding a grain or grains in the case of abundant, connected minerals like plagioclase and quartz. This type of
data is normally generated by proprietary automated mineralogy systems but could aid in geoscience disciplines
beyond ore geology or petroleum geology (Han et al., 2022). An illustrative example is in the analysis of grain-
scale properties of biotite. This is of wide interest because oxidation of ferrous Fe in biotite drives expansion of
biotite grains, which generates stresses in the surrounding rock that may be large enough to fracture the rock
(Fletcher et al., 2006; Goodfellow et al., 2016; Goodfellow and Hilley, 2022). To the extent that biotite expansion
promotes generation of regolith from bedrock, it may even influence the km-scale evolution of mountainous
topography (Wahrhaftig, 1965; Xu et al., 2022). In granitic rocks, numerical modelling has shown that biotite
abundance influences the accrual of microscale damage (Shen et al., 2019) and weathering profile development is
partially guided by biotite crystal size (Goodfellow and Hilley, 2022). These are two properties that can be
directly measured in our thin section-scale mineral maps.

To obtain such mineral maps in some previous studies, researchers have often engaged in manual or semi-
automated characterizations of sample mineral properties (Buss et al., 2008; Ündül, 2016). These workflows are
often tailored for a single study (e.g., Goodfellow et al., 2016). Methods that are based on generalizable
workflows involving automated mineralogy methods such as the one presented in this study could enhance
comparability between studies. Since we converted the predicted mineral maps into a vector (polygon) form
within QGIS, we could use built-in functions to gather large amounts of data on grain neighbours or perform
grain size measurements. For example, the 20 largest biotite grains in samples 1-1a and 6-3b comprise 80% and
94% of the total biotite area, respectively (Fig. 7a-b). The median grain area of these 20 biotite grains in sample
1-1a is 0.60 mm$^2$, several times larger than that in sample 6-3b (0.19 mm$^2$; Fig. 7c).

We can also use raster morphology operations on the mineral maps to measure distances between mineral phases.
In analog and numerical experiments that impose stress on granitic rocks (Tapponier and Brace, 1976; Li et al.,
2003; Mahboudi et al., 2012), biotite grains can act as preferential origination points for microfractures, but
biotite can also arrest propagation of microfractures arising from neighboring grains. Thus, the distance between
biotite grains may be an important, yet rarely measured property. In the example of the two samples in Fig. 7,
biotite grains have similar median distances from one another but different probability distributions of distances
between biotite grains, particularly in the long tail of the distributions at larger distances (Fig. 7e). We can also
extract the composition of neighbouring grains surrounding biotite (Fig. 7f), which reveal that chlorite is much
more abundant near biotite relative to the rest of the thin section. Data like these can be useful for those studying



the impacts of different grain-grain contacts on stress response during rock mechanics experiments (e.g., Aligholi
et al., 2019), which has shown that some mineral interactions can have an outsized influence on the development
of fractures and failure. In sum, the data in Fig. 7 illustrate the potential power of RF-generated mineral maps to
improve quantitative in-situ investigations of biotite weathering (Behrens et al., 2021) and form the basis for
more realistic models of biotite-driven rock damage (Shen et al., 2019).

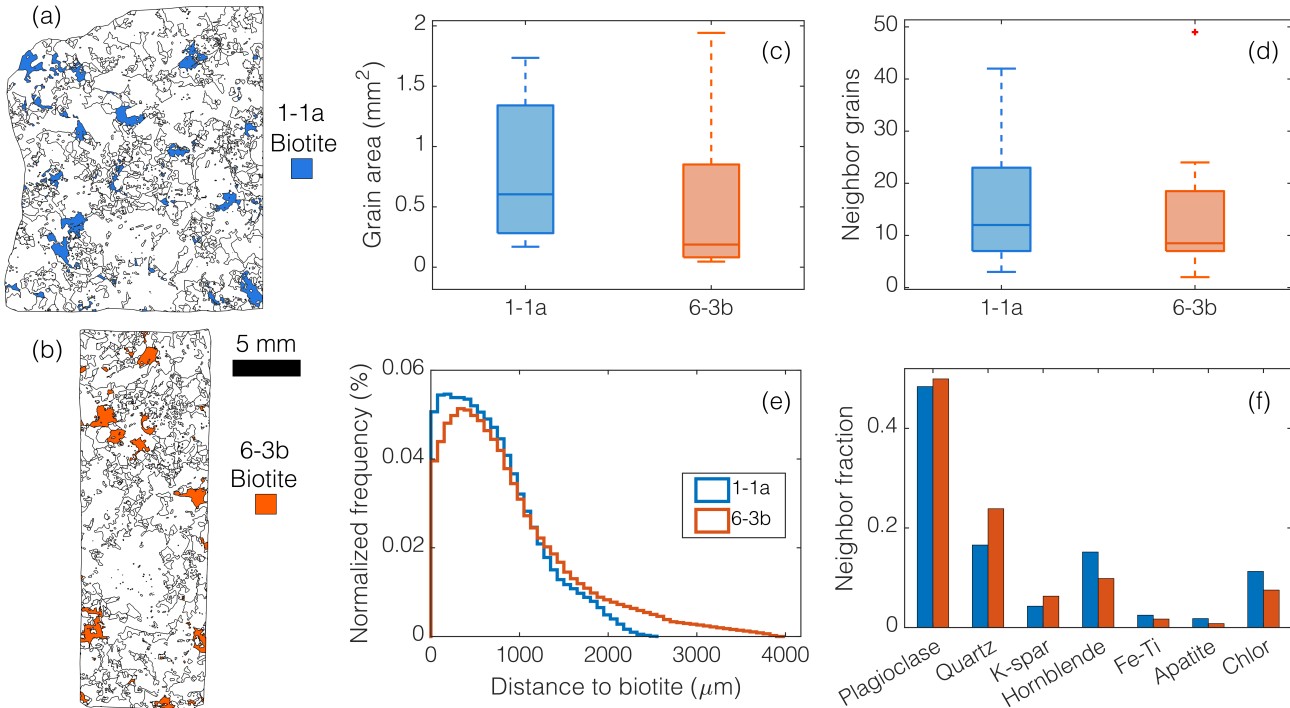

**Figure 7.** Example of quantities that can be obtained from mineral maps generated by the automated method in

this study. (a-b). Colours highlight biotite grains identified in the RF-generated mineral maps in thin sections 1-

1a (blue) and 6-3b (orange). (c-f). Biotite properties extracted from predicted maps for the 20 largest biotite

grains in each sample. These data could help inform numerical models of microcrack generation and allow for

quantitative comparisons between different samples or lithologies (e.g., Shen et al., 2019). (c) Boxplot of biotite

grain area (mm²) for the 20 largest biotite grains for both samples. (d) Boxplot of number of grains surrounding

the largest 20 biotite grains. (e) Normalized frequency distribution of distances between biotite pixels (not

including those inside a biotite grain). (f) Composition of neighbours as a fraction of perimeter.







### 5.3 Limitations

Our method's greatest asset is that it can generate thin section-scale mineral maps without requiring the use of propriety software or a background in programming. Its most important limitation is that it is most accurate if the user trains a RF model for every thin section sample. Using a RF model that was trained on one sample to predict mineral maps for another sample can yield mineral maps that accurately map mineral phases in some areas but inaccurately in others. For example, when we applied a RF model that was trained on sample 16-2a to sample 6-3a, apatite abundance was overpredicted by a factor of 5 possibly due to 6-3a having some highly calcic zones within plagioclase grains. So, for the most accurate results, we recommend training each thin section separately.

A second limitation is that this method tends to be less accurate at identifying low abundance phases. Unlike some proprietary automated mineralogy software systems, our method does not use predefined EDS spectra to identify mineral phases. Instead, our method trains RF models on the samples themselves, which means that each mineral phase of interest must be abundant enough to properly train the RF model. The relatively low F1 scores of the lower abundance phases in our samples (Table 2) suggest that the minimum abundance required to train a RF model is larger for minerals with small grain size (e.g., in the case of apatite) and a lack of compositional distinction (e.g., in the case of chlorite). Mineral phases must be resolvable by the EDS data, so collecting EDS data with a field-emission-gun SEM at higher resolution (~0.1 μm) could improve mineral classification in rocks with finer grain size distributions (Han et al., 2022).

A final limitation is that mineral grains that border mineral grains of the same phase appear to the RF model as regions of the same mineral, and hence can be classified as a single mineral grain, rather than two grains. This is a common issue shared with other automated mineralogy methods (Lanari et al., 2014; Hrtska et al., 2019), and it can affect inferred probability distributions of mineral grain size of those mineral phases if not properly accounted for.

### 6. Conclusions

The main contribution of this study is a new automated method for obtaining mineral maps from EDS scans of rock thin sections. This method is implemented within a free and open-source GIS application, uses free and open-source plugins for RF image classification, and requires no programming. To demonstrate the utility of this method, we trained RF models on EDS scans of 14 thin-section samples of a well-studied, plutonic igneous rock. The resulting model-predicted mineral maps compare well with manually delineated mineralogy maps, with 95%



of pixels on the mineral maps predicted correctly. With regards to the most abundant minerals in the Rio Blanco
tonalite, plagioclase feldspar and quartz, the models attained 96% and 94% accuracy, respectively.

We utilized scikit-learn's implementation of the RF classifier to search for optimal RF hyperparameters and to
test input feature (element) importance. We saw no increase in accuracy using optimal hyperparameters found in
scikit-learn when used within Orfeo Toolbox, so we recommend using the default hyperparameters. We did see
that an important input feature, K, did lower accuracy when not included in Orfeo Toolbox-based models, so
some level of experimentation with input features during the training step is warranted. We also tested to see if
our pre- and post-processing steps had a large influence on accuracy by using different sizes of mean and
majority filters. An absence of filtering and excessively large filters led to lower accuracy while filters in the
range of 5-10 pixels for both mean and majority filters led to higher accuracy.

Situating the workflow within a free and open-source GIS environment confers distinct advantages. Open source
extends benefits such as source code availability, extensive documentation, and accessibility. Moreover, as the
workflow is within a GIS environment, the application is likely to be familiar to a range of geoscientists. Also, all
the available tools (e.g., different types of image filters) within the GIS allow for easy input feature
experimentation. The mineral maps from our method proved highly accurate when compared to manually-
delineated maps, and estimates of mineral abundance compared well to previous estimates from the literature for
our sample lithology. Many of the measured quantities produced by proprietary automated mineralogy systems
are obtainable once predicted mineral maps are converted to vector datasets. These measurements, such as
median grain size and amount of grain neighbours, can be useful to researchers studying microscale damage
processes that arise through rock weathering or rock mechanics experiments. We hope that this method will be
useful for researchers who wish to obtain rapid, automated mineralogy maps of thin sections.

**Code and Data availability**
The manuscript supplement containing the code for analysis and visualizations is available through a Zenodo
repository (https://zenodo.org/doi/10.5281/zenodo.10912627; Reed et al., 2024). The supplement also contains
data (smoothed elemental intensity rasters, training polygons, and test maps) for the three thin sections with
manually delineated test maps.

**Author contribution**



**Miles Reed**: conceptualization, formal analysis, methodology, software, visualization, and writing (original draft and preparation); **Ken Ferrier**: funding acquisition, supervision, visualization, and writing (review and editing); **William Nachlas**: resources and writing (review and editing); **Bil Schneider**: investigation and writing (review and editing); **Chloe Arson**: funding acquisition and writing (review and editing); **Tingting Xu**: writing (review and editing); **Xianda Shen**: writing (review and editing); **Nicole West**: funding acquisition and writing (review and editing).

**Competing interests**

The authors declare no competing interests.

**Acknowledgements**

This work was supported by NSF award EAR-1934458 and NSF award EAR-1755321.

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
