# Peer review of "A free, open-source method for automated mapping of quantitative mineralogy from energy-dispersive X- ray spectroscopy scans of rock thin sections"

_EGUsphere, 2024_

## Author Response (AR1)

We thank the reviewers and editor for their insightful comments and helpful feedback. In this document we list *reviewer comments in italics* and we list our responses to all comments in normal font.

Review #1

**General Comments**

*The manuscript describes the analysis of 14 tonalite thin sections by SEM-EDS mapping and demonstrates an approach of mineral identification by the random forest classifier. The results are analyzed for the modal mineralogy and some geometric properties of the samples. The focus is on the approach, which was realized through open-source software only, giving access to any user of such data without the need of programming skills or expensive software. The need for and advantages of free software are an important topic within the community. High prices and restrictions of proprietary software are often a problem. Therefore, developing and providing the access to such kind of analysis is an important contribution to the scientific community and the readers of the journal.*

*In general, the results of the manuscript are convincing and the conclusions sound. The findings are well presented. The discussion might need some additions (see specific comments below). The structure, however, seemed a bit problematic, as there is some redundancy or repetition in the general approach and the used method. The reader has to move a lot in the manuscript when a topic is split into four different chapters. The reader would benefit from a condensation of the description.*

*I would suggest making a shorter version of the general approach (current chapter 2) with no discussion or recommendations (l. 134f for the procedure), but just the plain and neutral description of the method. Then in chapter 3 explain in more detail why certain parameters are important and were used in this study. Then the discussion should mention further explanations and limitations of the methods. Otherwise, there is in parts no real separation of results and discussion/interpretation.*

*As a personal preference: The style of the manuscript is in parts similar to a user manual. For a scientific paper, I would suggest to aim at a more neutral style of description in the results part.*

*In general, this work is useful research and should be published after the comments have been addressed.*

**General response:**

Thank you for your positive and thorough review of our manuscript. We agree with your suggestion to keep Section 2 brief, and we have made minor edits to this section to trim its length accordingly. We have also made additional edits to the text in various places to aim toward a more neutral tone, as you recommend (see responses to specific comments below).

We have retained the general format of the manuscript, as we believe this structure will be helpful for readers who would like to assess the utility of this method for their own work. To emphasize that Section 4 focuses on discussion of testing of inputs and model hyperparameters that go beyond the results of the case study in Section 3, we retitled Section 4 'Discussion: Accuracy of random forest-predicted mineral maps and sensitivity analyses', and we retitled Section 5 'Discussion: Advantages, utility, and limitations'. We made several edits to the text to add clarifying context, as suggested (see our comments to specific comments below). Altogether, we believe these edits will help guide readers through the manuscript more effectively, and we thank the reviewer for the recommendations.

Below, we respond to each specific comment individually.

***Specific comments:***

*L. 28: Do you consider hyperparameter a generally known expression? If not, it would be better to use a more general word in the abstract.*

In machine learning, hyperparameter is a well-defined term in widespread usage. To define this more clearly in the text, we added the parenthetical "(i.e., tuneable settings that control model behaviour)" to **Line 26**, the first place it appears in the abstract, and **Line 282**, the first place it appears in the main text.

*Ll. 39f: I would suggest quoting the original papers regarding quantitative mineralogy, e.g.:*

• *Sutherland, D. N. and Gottlieb, P.: Application of automated quantitative mineralogy in mineral processing, Miner. Eng., 4, 753–762, 1991.*

• *Sutherland, D., P. Gottlieb, R. Jackson, G. Wilkie und P. Stewart (1988). "Measurement in section of particles of known composition." Minerals Engineering 1(4): 317-326.*

• *Gu, Y.: Automated scanning electron microscope based mineral liberation analysis An introduction to JKMRC/FEI mineral liberation analyser, Journal of Minerals and Materials Characterization and Engineering, 2, 33, 2003*

Thank you for the suggestion. As suggested, we added citations to these canonical papers and added 'automated' and 'computerized' to **Lines 38-40**.

*L. 52f: I am not aware of a WDS system that can map mineral phases. I would be interested in more information.*

We had included WDS in this sentence as a possible input to automated mineralogy systems because the cited reference (now removed from that location), Ortolano et al. (2018), mentioned that it can be a potential main input to their system. However, they only used spot results from WDS as a way to calibrate their usage of EDS data, and elsewhere in the literature, it does not appear that WDS has ever been used due to logistical concerns like time cost. We have therefore removed mention of WDS from **Line 53**.

*L. 53f: I am not sure about LA-ICP-MS, here. From the quotation, you might mean Laser Induced Breakdown Spectroscopy (LIBS).*

Thank you for catching this. We changed this to LIBS on **Line 54**.

*L. 77: "predicted mineralogy". You cannot predict mineralogy. What you mean is the modal mineralogy or mineral abundance.*

As suggested, we corrected this to 'modal mineralogy' in **Line 78**. We also changed 'RF-predicted mineralogy' to 'RF-predicted minerals' in **Line 292**.

*Ll. 88ff: The last paragraph of the introduction is normally a short summary of the hypothesis or goal of the paper and how it might be reached. You start with the goal in l. 88, but then continue to explain classification algorithms. Maybe you could move the algorithm part up and leave the goals to the last paragraph.*

Thank you for the suggestion. We revised the last paragraph of the introduction to now more plainly state the goal of paper. This entailed moving the goal statement (previously Lines 88-89) and the statement on our usage of Orfeo Toolbox (previously Line 91-93) to **Lines 103-104** and **106-108**, respectively. We also

removed a sentence at the beginning of the previous paragraph to make it more coherent with regards to random forest classification.

*L. 96. "random sampling": Do you mean a random sample?*

Yes. We changed this to 'random sample on **Line 92**.

*L. 110. "In the remainder of this study": Better: Furthermore, ...*

We changed this to 'furthermore'.

*L. 113. "hope": You could write that you intend to make it available to a broader community, which is good, but hope is the wrong word in this context. An ideal scientist would be neutral. The same applies to l.614.*

We changed "hope this method provides" to "intend to provide" on **Line 112**.

*L. 118: You should clarify that you mean EDS data from a SEM. There are other methods producing EDS spectra, some of which you list in the introduction, so until you mention it, the reader assumes EDS data from any source.*

We added "acquired using a SEM" to **Line 118**, as suggested.

*L. 127: How do you deal with overlapping peaks (e.g. S/As/Pb or Ti/Ba etc.)? What influence would overlaps have on the classification? Would it be possible to have several lines (e.g. K-alpha, K-beta, L-alpha) for one element?*

These are interesting questions, but they are beyond the scope of this study. To clarify this, we edited **Lines 123** to read: "The starting point for this method is elemental rasters derived from EDS-generated scans of rock thin sections." The sentence after this one notes that "for the purposes of this method, we take these scans as already measured and in hand." Since the purpose of our study is to use elemental rasters to generate mineral maps, not to investigate the effects of overlapping peaks or elemental rasters from multiple X-ray, we do not address those issues here.

*Ll. 132ff: This raises several questions: If you need to know the minerals already, how would you do that? This would mean a lot of work done twice (manual identification + RF classification). What happens, if you miss one or more minerals? Can you analyze unknown samples?*

**Lines 132-134** list potential methods for identifying minerals present in samples: "This also involves compiling a list of all the mineral phases that will be mapped in the thin section, which can be assessed based on prior knowledge, literature, and examination of EDS spectra". In practice this is relatively straightforward, since it involves only identifying which minerals are present, not quantifying their abundance or mapping their spatial arrangement.

If minerals are missed in the training data, the classification would be less accurate to the extent that any missed minerals constitute a significant portion of the sample, which is why we recommend that users include all minerals present at abundances > 0.1% in the training (see our response below to the comment at Ll. 164ff). To emphasize that users should take care to identify all major minerals present in their samples, we added the following in **Lines 267-270:** "We note that during this training step, the user should take care not to misidentify or neglect training upon abundant minerals, which could have a

detrimental effect on the classification accuracy. To prevent this outcome, we used all available elemental rasters to verify that training polygons were within the bounds of the identified mineral."

Although a small amount of work is done twice, the upside of our method is that mineral maps are generated through the relatively simple process of training random forest models, which takes far less time (and may be more accurate) than hand digitizing a detailed mineral map of an entire thin section. Thus, the potential payoff for the initial effort identifying minerals is large.

Upon running EDS and examining a sample under microscope to identify minerals, a sample would cease to be unknown, and, thus, usable in our method. In **Lines 134-136**, we emphasize that this method is not for completely unknown samples that resist all efforts to identify minerals: "Our method is not viable for those thin sections from completely unknown lithologies that resist efforts to identify minerals under the microscope and/or manual examination of EDS data.".

*L. 138 "electron beam": It could also be X-rays or laser from what you mentioned in the introduction. Consider also the comment for l. 118.*

Since this sentence refers only to EDS, not other methods (e.g., X-ray or laser-based methods), we have left "electron beam" in this sentence as is.

*L. 139: How would the smoothing help, when a peak was identified incorrectly? Please clarify.*

Thank you for the suggestion. To avoid giving readers the impression that smoothing corrects for misidentified peaks, we removed the reference to peak misidentification here.

*L. 158: I believe 30% training data would be standard. How did you select the training areas? Randomly?*

In the literature, there appears to be no universally applied standard in the proportion of the total data allocated to training data. Instead, a common rule of thumb is that hundreds to thousands of training samples are needed in RF (Cutler et al., 2012). In the example we present in our manuscript, we exceeded this criterion dramatically as any pixel within a training polygon becomes a training sample. To clarify this and to give users a target to aim for, we modified the end of the sentence at **Lines 157-159** to 'we recommend restricting the location of these small training polygons to a relatively small portion of the thin section (~10-20% by area)'. We also added the following to **Lines 272-273** to give readers context for standard practices: "Hundreds to thousands of pixel-level training samples per class are generally considered sufficient for RF models (Cutler et al., 2012)."

We selected training areas by 1) identifying areas that included all minerals, and 2) attempting to maximize the amount of available training samples for each mineral. We mention this in **Lines 256-257**: "We selected training areas that represented all minerals as well as possible, so that each mineral would receive an adequate amount of training data for each mineral."

*L. 165 expression "mineral phase": phase is a defined technical term. The algorithm assigns a mineral name to a pixel, since EDS contain have only chemical information, which cannot always distinguish phases (polymorphs, limitations in element analysis such as light elements).*

To avoid this potential ambiguity, we now use the term 'mineral class', since we are conducting a random forest image classification. We also have replaced 'mineral phase' elsewhere in the manuscript in favor of 'mineral' or 'mineral class'.

*Ll. 164ff: What about minerals that have not been trained? Are all pixels classified? Can the similarity between training pixel and the classified pixel be described?*

Minerals that have not been trained upon are classified as one or more of the trained minerals. As we describe in our response below to L. 239, we recommend training all minerals except those present in trace abundances (< 0.1%).

Almost all pixels are classified. The rare exceptions are the ~0.0005% of pixels along mineral edges that are identified as "ties", in which the underlying trees in the random forest arrive at different classifications with no clear majority winner. These tie pixels are not classified but marked as NaNs (not a number) in the output. These NaNs are then smoothed over by the majority filter in the post-processing step (**Lines 173-174**).

Yes, the similarity between training pixels and classified pixel can be described. At **Line 382-387**, we present an example of this, describing how the combined confusion matrices yield comparisons between the test and predicted maps.

*Ll. 173f: How did you find that 10 pixels are best? I assume this would vary, based on the sample type, grain size and resolution of the image. Can you offer an explanation, why pixel sizes influence accuracy? I would assume that, if you have less spatial blur (mean filter), the results would be more accurate. Could it be a user bias, as users prefer large grains for training areas?*

We provide an answer to this question in Section 4.3, where we describe how we found that a 10-pixel filter worked best on our example samples. **Line 171** states that this will be discussed later in the manuscript: "As we describe in Section 4.3, we found that this was best done with a 10-pixel radius majority filter in the example samples we applied this to".

Why a filter window size works best on a given thin section could be a complex mix of factors, as you assume. This is why we emphasize at **Lines 489-491** that users should test window sizes (something that is easy to do within QGIS) if F1 scores are low during training: "These optimal sizes most likely result from a mix of the initial EDS pixel resolution and data quality and the types and sizes of minerals in the thin section (Lanari et al., 2014; Ortolano et al., 2018), so we recommend that users experiment to find the optimum filter sizes for their samples."

The individual training polygons can be any size, so small mineral grains can be trained upon just as easily as large mineral grains. We therefore don't expect that this would result in user bias.

*L. 175: You should explain voting ties and how they are treated in the introduction.*

Please see our discussion of ties in our response to the comment on "Ll. 164ff" above.

*L. 221: How can a beam step size and magnification result in a map? Please rephrase. Also, is beam step size the same as pixel size?*

To clarify this, we altered **Lines 209-211** to: "For the measurements on our thin sections, the instrument and accompanying software produced full thin-section elemental intensity maps (counts/eV) at a resolution of 4 μm/pixel, which was determined by the beam step size."

Yes, beam step size directly determines the output raster resolution (pixel size).

*Chapter 3.2: If you have a mean raster with 20'000'000 pixels and measure 200 ms per pixel, how can you calculate an acquisition time of 3.5 h? Please explain.*

Thank you for catching this—this was our mistake. In Section 3.2 of the previous version of the manuscript, the stated acquisition time (3.5 hours) was correct, but the stated dwell time of 200 ms was incorrect. The actual dwell times were, on average, several orders of magnitude lower than this, and varied during the scans. In practice, the acquisition time is governed by the 'process time' setting in the EDS pulse processor (also known as time constant by some manufacturers) in conjunction with the dynamically controlled dead time interval. Rather than stating the dwell time, it is conventional to report the process time, so we removed the sentence about dwell time and added the following sentence about process time at **Lines 212-214**. "EDS process time (also known as 'time constant' by some manufacturers) was 4, which is an intermediate value that balances acquisition time and data quality."

*L. 239: Why this recommendation? What if apatite is important to a study? Why would you combine minerals? One would assume that this affects accuracy negatively. Do you have any information on how many training areas are necessary and how large they should be? You should have a discussion on the selection of training areas in the discussion section.*

We recommend omitting trace mineral phases (i.e., those with abundances < 0.1%) because they don't have enough mineral area to train upon and hence would be challenging to map accurately. Because such minerals constitute a very small portion of the sample, omitting them has only a small effect on the inferred mineral maps. To clarify this, we edited **Line 238-240** as follows. "We recommend that minerals present at abundances lower than this be omitted or combined with the understanding that overall accuracy is most likely being negatively impacted in a minor way." Earlier in the manuscript, we emphasized that this method is not suitable for users interested in trace minerals (**Lines 136-137**): "For those workers that require high accuracy in very low abundance minerals, our method is not advisable."

Section 3.4 details how we selected training areas. Please also see our response to the comment on "L. 158" above.

*L. 246f: Move the explanation of virtual rasters to the overview of the method or the introduction.*

Thank you for the suggestion. As suggested, we added "a type of container for multiple rasters" to Lines **147-148** to give a brief description of what a virtual raster is for clarity.

*Figure 2: Could you plot the data from XRD and point counting as well for comparison? Then you could shorten the paragraph above.*

This is a great recommendation. We added the data points to Figure 2, as suggested, but retained the text for completeness.

*L. 337: Why did you choose these three thin sections?*

We selected these thin sections because they appeared to be representative of the larger group of thin sections. To clarify this, we edited **Lines 341-342** to read: "we manually mapped the mineralogy of a small section for three representative samples".

*Ll. 363f "This indicates that the models correctly predicted apatite when attempted but the models often neglected to predict apatite.": Why would the models not try to predict apatite, if they are trained for apatite? Please explain.*

To explain why the models occasionally do not try to predict apatite, we added the following to **Lines 368-372**: "Because apatite is rare and appears as small inclusions in our samples, less training data was

collected for it than for other minerals in each sample. This can result in class imbalances in training data, which, for rare mineral classes (in our case, apatite), can produce scenarios in which the model does not try to predict the mineral class, as the diversity of training data for rare classes remains relatively low (He and Garcia, 2009)."

*L. 356 and Ll. 391f: You use questions to structure your manuscript. I would suggest avoiding that and use a more descriptive style.*

We changed these questions to descriptive statements, as suggested.

*Table 3: You could add a line with the standard parameters for comparison. If I understood correctly, you used the standard values for what you showed in figure 2?*

There are no true 'standard' values for random-forest classification hyperparameters (Probst et al., 2019), but we now include in Table 1 the reasonable ranges from Probst et al. (2019). We used the default values in Orfeo Toolbox's implementation of random forest classification. We edited **Lines 285-286** as follows to indicate that, here, 'default' means values pre-selected in Orfeo Toolbox. "We used the default hyperparameter values pre-selected in Orfeo Toolbox (Table 1) for the models employed for our final predicted mineral maps."

*Ll. 535-543: How do you evaluate the effort to label training data compared to what you describe? In how far can the training data be extrapolated to unknown samples or other studies? As the training requires most of the work and time, what are the limits of this approach? – Some of the questions are answered further in the manuscript, which raises the question, if the text could be restructured.*

The training-data labeling process used in our method is straightforward and relatively quick, as all the pixels within a training polygon are labeled simultaneously. Multiple thin sections could easily be trained in a single day. We are unable to compare the effort required to implement our method to that required for the manual or semi-automated methods in those studies since we did not attempt to replicate their methods.

Training data from one of our thin sections does not extrapolate well to others, so it would not be useful on unknown samples that resist mineral identification. At **Lines 576-577** we highlight that a limitation of the approach with regards to model training is that every thin section needs trained to achieve the highest accuracy, so applications encompassing many hundreds of thin sections would prove infeasible.

As we describe in our response to the reviewer's General Comments above, we prefer the structure of the manuscript as it is, so we have kept the general structure of the manuscript intact.

*Ll.545-558: How can you make sure that you actually have identified only one grain of e.g. biotite? With EDX, you get the chemical distribution, but if you have monomineralic aggregates or touching biotite grains, how can you separate those grains in order to calculate correct grain areas/ sizes? – After reading chapter 5.3, I realized that you addressed part of the comments. The manuscript would benefit in structure and clarity, if you could combine the paragraphs or address the problems/limitations in the discussion parts directly where you mention the analysis.*

Very true, and, as you note, we discuss this later on in Section 5.3. To point this out to readers at this place in the manuscript, we added the following to **Lines 549-552**: "As we discuss in Section 5.3, classified biotite 'grains' may contain multiple bordering crystals of the same mineral as the EDS input data and the resultant classification cannot differentiate boundaries by elements alone (Lanari et al., 2014). As biotites

are relatively isolated from each other in our thin sections, these measurements serve as a reasonable indicator of true biotite properties."

We have left Section 5.3 unchanged as it serves to address these and other concerns in one place.

*General Discussion: Do you have information on the time invested in creating and testing the models? Can you put in relation how much time the mineralogical analysis (e.g. XRD) costs compared to your approach?*

Training the models is quick once the EDS elemental data is in hand and the minerals that need to be trained upon are identified. As we note in our response to L. 535-543 above, several thin sections can be trained in a day. By contrast, testing the accuracy of models took much longer because it involved creating manually-delineated mineral maps, which is the most time consuming step in the method.

Conventional powder XRD only provides a sample's modal mineralogy, which eliminates any spatial information about the minerals (e.g., size, shape, proximity). In this regard, it provides zero-dimensional information about the sample, so it isn't directly comparable to the 2D mineral maps generated in our method. We therefore have not added a comparison between this method and XRD here.

Technical corrections:

*L. 108 "requires no programming on the part of the user": This sounds a bit odd. Do you mean no programming by the user?*

We changed this to "by the user", as suggested (**Line 106**).

*L. 177: Can a map be interrogated?*

This phrasing is sometimes used in GIS but it does carry a level of unnecessary personification. We changed 'interrogated' to 'examined' on **Line 176**.

*L. 219: "studied" instead of "study"?*

We changed 'study' to 'studied' on **Line 219**, as suggested.

*L. 500: comma is unnecessary.*

Comma removed, as suggested.

Review #2

*The authors should address the issue of the real added value of their work with respect to the pertinent literature.*

Thank you for the suggestion. The added value of this work lies in the development of a new, freely available, user-friendly method to obtain quantitative mineralogy maps from EDS scans of rock thin sections. This is described in multiple areas of the manuscript. Section 1 describes the context of the pertinent literature on the limitations of commercial approaches for automated mineralogy (**Lines 77-88**) and the advantages of situating our method within QGIS, which facilitates application of the method for more users (**Lines 103-113**). In Section 5.1, **Lines 498-525** describe the key advantages of our method over alternative methods, including: any worker with EDS elemental intensity maps can use our method;

this method is built upon freely available software that is maintained by very large user bases; and users do not need any programming knowledge to use our method. These three things add a great deal of value.

*It is not clear the assumption at the basis of considering the results used for the training on a region as useful for a test area.*

Thank you for the suggestion. This is, in fact, already addressed in detail in the heart of the manuscript. Section 3.4 describes how the random forest models are trained on a region, Section 3.5 describes how the trained models are applied to other regions of the thin section, and Section 4.1 describes the accuracy of the resulting mineral maps in test areas. Figures 3 and 4 and Table 2 show that model-predicted mineral maps are highly accurate relative to the test maps. Thus, these results support the assumption that training the model on a given region of the thin section and applying it to other regions of the thin section yields accurate results.

Comment #1

*The text is well written and organised. The limits of the method are given. The authors tested their approach to one of the most simple geological objects, an undeformed granite. Apparently it seems to work by producing a spectral map. However, this approach to SEM-AM is of very limited use for application to many mineral processing issues when compared to the professional software platforms, e.g. MLA 3.1 by FEI.*

Thank you for your comment. The goal of this method is to present a free, open-source way to create detailed mineral maps from widely accessible EDS scans, rather than presenting a software platform that would duplicate all of the features offered by professional software platforms. These mineral maps can be used by workers conducting research on chemical weathering or rock mechanics for whom these mineral maps may suffice for their needs.

*The authors should provide a further schematic Figure which explains how the primary EDS signal is transferred to the element map, see more detailed comment in the text.*

As we describe in our response to Reviewer #1, the purpose of our study is to use elemental rasters to generate mineral maps, not to investigate how EDS signals are converted to elemental rasters, so we do not address that here.